# Tissue Adaptation to Environmental Cues by Symmetric and Asymmetric Division Modes of Intestinal Stem Cells

**DOI:** 10.3390/ijms21176362

**Published:** 2020-09-02

**Authors:** Aurélia Joly, Raphaël Rousset

**Affiliations:** Université Côte d’Azur, CNRS, INRAE, ISA, 06903 Sophia Antipolis, France; aurelia.joly@etu.univ-cotedazur.fr

**Keywords:** stem cell, symmetric/asymmetric divisions, gut, JNK, *Drosophila*, mouse, adaptive growth, regeneration, stress, aging

## Abstract

Tissues must adapt to the different external stimuli so that organisms can survive in their environments. The intestine is a vital organ involved in food processing and absorption, as well as in innate immune response. Its adaptation to environmental cues such as diet and biotic/abiotic stress involves regulation of the proliferative rate and a switch of division mode (asymmetric versus symmetric) of intestinal stem cells (ISC). In this review, we outline the current comprehension of the physiological and molecular mechanisms implicated in stem cell division modes in the adult *Drosophila* midgut. We present the signaling pathways and polarity cues that control the mitotic spindle orientation, which is the terminal determinant ensuring execution of the division mode. We review these events during gut homeostasis, as well as during its response to nutrient availability, bacterial infection, chemical damage, and aging. JNK signaling acts as a central player, being involved in each of these conditions as a direct regulator of spindle orientation. The studies of the mechanisms regulating ISC divisions allow a better understanding of how adult stem cells integrate different signals to control tissue plasticity, and of how various diseases, notably cancers, arise from their alterations.

## 1. Introduction

In contact with ingested food, drink, and medications, the intestinal epithelium is continuously subjected to various aggressions such as bacteria, viruses, or chemical compounds. To keep its integrity, the intestinal epithelium is in constant renewal, replacing old, injured, or dead cells. This renewal is ensured by multipotent intestinal stem cells (ISC), of somatic origin and present within the tissue itself, which adapt their proliferation to daily needs (homeostasis, adaptive growth) or during aggression (regeneration). In the mouse small intestine, these ISC express the R-spondin receptor Lgr5 (Leucine Rich Repeat Containing G Protein-Coupled Receptor 5) and are located at the base of the crypts (Figure 1a). Besides these active ISC, another type of ISC located at the +4 position relative to the crypt base is present. The ISC +4 are quiescent during normal homeostasis, and their exit from quiescence occurs when the active ISC are compromised, therefore constituting reserve ISC [1,2]. Despite their stem cell potential, the ISC +4 are precursors committed to becoming secretory cells, revealing the plasticity of this cell type [3,4]. During the differentiation process, cells progress from the crypt to the villi to first become transit-amplifying precursors (TA) that retain the ability to divide (Figure 1a). Absorptive precursors then differentiate into enterocytes (EC), whereas secretory precursors can give rise to enteroendocrine cells (EEC; secreting hormones), goblet cells (secreting mucus), tuft cells (secreting interleukins), and Paneth cells (secreting anti-microbial peptides and niche factors). Active cell migration relying on actin-rich basal protrusions, rather than passive migration, has recently been shown to allow epithelial cells to migrate upward to the villi [5].

The *Drosophila* midgut, which is the counterpart of the mouse small intestine, is a well-recognized model to study tissue renewal and plasticity during homeostasis, regeneration, and aging. Despite some physiological divergence, the cell lineage and molecular mechanisms involved in ISC proliferation and differentiation are well conserved between *Drosophila* and the mouse (Figure 1) [6]. Recently, single-cell RNA sequencing advances allowed the complete description of the different cell types of the mouse small intestine and the *Drosophila* midgut [7,8]. These studies confirmed the similarities between the two tissues, each containing stem cells, precursors, enterocytes, and enteroendocrine cells (Figure 1). The mouse goblet, tuft, and Paneth cells are absent from the fly intestinal epithelium, their specific functions being assumed by the enterocytes and enteroendocrine cells. Indeed, based on marker gene expression, *Drosophila* enteroendocrine cells are similar to mouse enteroendocrine cells, goblet cells, and tuft cells. *Drosophila* cardia cells, localized in the anterior midgut extremity and known to secrete components of the peritrophic membrane, which has the same protective role as the mammalian mucus, are also equivalent to goblet cells [8]. In addition, Paneth cells are related to lysozymes-producing anterior enterocytes. Interestingly, five *Drosophila* cell types could not be clearly specified, although three of them could be part of the enterocyte lineage [8]. Assigning markers for each of them will help attribute a function to these new cell types of the *Drosophila* midgut.

In *Drosophila*, only active ISC have been described so far, located at the basal side of the gut epithelium and expressing the transcription factor Escargot (Esg) and the Notch (N) ligand Delta (Dl) [9,10]. It was first proposed that ISC divide asymmetrically to both self-renew and give rise to a unique precursor, called enteroblast (EB), expressing the transcriptional reporter of Notch activity, Su(H)GBE (Suppressor of Hairless and Grainyhead Binding Elements) (Figure 2). Three studies then showed that enteroendocrine cells come from a different precursor, Su(H)GBE negative and Esg-Dl-Prospero (Pros) positive, called enteroendocrine precursor (EEP) [11,12,13]. Later, the first, and so far the only specific enteroendocrine precursor marker, named Piezo, was discovered [14]. Piezo-expressing cells are also marked by Esg and Dl, but not all the Piezo-positive cells express Pros, revealing two different enteroendocrine precursor cell types [14,15]. Careful analysis of the enteroendocrine lineage is now required to highlight the various steps of differentiation from the ISC to the enteroendocrine cells. Different types of enteroendocrine cells have been described in the midgut, depending on the cocktails of hormones they produce [16,17]. Recently, single-cell analyses revealed the precise composition and distribution of the different types of enteroendocrine cells along the midgut [8,18]. Ten different subtypes, each of them expressing between two to five gut hormones, were discovered and grouped in two major classes: the Allatostatin C-expressing class I and the Tachykinin expressing class II (Figure 2). After ISC asymmetric division, the resulting enteroendocrine precursor can directly differentiate into the enteroendocrine cell, but in 71% of cases, it divides to give rise to two enteroendocrine cells, indicating that ISC are not the only cell type to divide in the midgut [19]. However, the Piezo-positive enteroendocrine precursors have a reduced capacity of proliferation compared to Piezo-negative ISC [14]. Enteroendocrine precursor division is itself asymmetric, forming one class I enteroendocrine cell and one class II enteroendocrine cell, the latter being specified by N signaling [13,16,18,19] (Figure 2). Finally, enteroendocrine cells can also be produced via symmetric ISC divisions or by direct differentiation of the ISC [11,12], although the latter process has later been challenged [19].

By adapting their activity to the need, the ISC are the mainstay of homeostasis and regeneration of the tissue. To do this, they respond to mechanical and molecular signals coming from numerous signaling pathways (Notch, TGFβ/BMP, Hpo/Yki, EGFR, JNK, JAK/STAT, Wnt/Wg, PVR, IIS, TOR, Hh, Integrins), modulating their proliferation and differentiation rates. An important adaptive mechanism is based on the switch between the division modes of ISC, asymmetric versus symmetric. Whereas the asymmetric division generates two daughter cells of different fate, one self-renewed ISC, and one precursor (enteroblast or enteroendocrine precursor) as seen above, the symmetric division gives rise to two ISC (symmetric self-renewal) or two precursors (symmetric differentiation) (Figure 2). Although both asymmetric and symmetric divisions occur during homeostasis, the switch between the two modes is particularly important for the intestine to adapt to different conditions, such as nutrient availability, external aggression, or aging. The mechanisms involved in this regulation have started to be better described, though much work needs to be done for a complete understanding of the molecular processes controlling the adaptive capacity of adult stem cells of the intestine.

## 2. Asymmetric Versus Symmetric Divisions during Homeostasis

In the developing mouse intestinal crypt, Lgr5 + ISC adopt asymmetric divisions to generate precursors only after all ISC-ISC symmetric divisions have taken place in order to reach the necessary number of cells for proper tissue growth [20,21]. Later, in the small intestine of adult mice, the majority of Lgr5 + ISC divisions are symmetric, giving rise to two ISC or two fast-lived TA [22,23,24]. This main mode of division therefore leads to a neutral drift of the crypts, whereby progressive loss of ISC, when two TA are formed, is compensated by clonal expansion of the duplicated ISC. However, this model has recently been challenged by a study showing that the neutral drift is rather dominated by asymmetric cell division, which represents 60% of ISC division during normal homeostasis [25]. It is likely that this discrepancy results from the difference in food quality used in the two studies (see below).

During fly gut homeostasis, approximately 80% of ISC undergo asymmetric division (ISC-enteroblast), whereas only 20% undergo symmetric division, leading to self-renewing (ISC-ISC) or differentiation (enteroblast-enteroblast) [26,27,28,29,30,31,32,33,34] (Figure 2). Clonal analysis revealed that, whereas the majority of multicellular clones contain one Dl-positive ISC, the other clones have several ISC, or none [26]. Therefore, the asymmetric fate produces enterocytes and enteroendocrine cells, while the symmetric fate leads to a neutral competition at the population level.

### 2.1. Delta/Notch Signaling Pathway

The Notch pathway is an evolutionarily conserved regulator of stem cell maintenance [35]. The signaling cascade is activated when one of the transmembrane Notch ligands Dl or Serrate (Ser), present at the surface of a cell, binds to its receptor Notch exposed by a neighboring cell. This induces cleavage of the Notch intracellular domain that enters the nucleus and transcriptionally regulates gene expression along with co-factors such as Suppressor of Hairless (Su(H)). In dividing ISC, the decision of daughter cells to self-renew or differentiate is dictated by the Notch pathway, whose high level of activation in enteroblasts prevents ISC fate and promotes differentiation [32,33,36]. Dl is the unique ligand in this process, as loss of Ser has no effect on ISC maintenance and enteroblast production [36]. Initially at the anaphase stage, Dl segregates equally in the two dividing daughter cells [36]. Later, during cytokinesis, Notch and Dl were shown to be asymmetrically dispatched in enteroblasts in a process depending on Sara endosomes, thereby promoting asymmetric division [37] (Figure 2). Interestingly, Notch signaling acts bidirectionally and is important not only for enteroblast differentiation, but also for the enteroendocrine lineage [13]. In the enterocyte lineage, ISC (high Dl) remain attached to the basement membrane, whereas enteroblasts (high Notch signaling) are more apical [9,10,13,36]. In contrast, in the enteroendocrine lineage, enteroendocrine precursor daughter cells are localized basally and express low levels of Dl, inducing weak Notch signaling in the ISC [13]. This important work undertaken in pupal guts, as well as in adult guts, revealed that Notch signaling is bidirectional with respect to the enterocyte and enteroendocrine lineages, but it is unidirectional with respect to the basement membrane, as basally located cells activate cells that are more apical, and not vice versa [13]. It is interesting to note, therefore, that two types of ISC co-exist in the *Drosophila* midgut, the Esg/Dl-positive ISC and the Esg-positive/Dl-negative ISC, although the latter are certainly transient and quickly re-express the Dl protein. Notch loss of function induces mosaic tumors made of ISC-like and enteroendocrine-like cells [36], and the bidirectionality of Notch signaling can explain this phenotype: loss of Notch leads to an accumulation of Dl-positive ISC in the enterocyte differentiation pathway and Pros-positive cells in the enteroendocrine differentiation pathway. However, activation of Notch signaling in the enteroendocrine lineage could not be confirmed using a Notch activity reporter generated by tagging the intracellular domain of Notch with GFP (NiGFP), as it was found in an inactive state at the plasma membrane [33]. One simple explanation is that NiGFP is not sensitive enough to detect the low level of Notch activity described in the enteroendocrine lineage. This study importantly identified another regulator of the choice between the enterocyte and enteroendocrine lineages, the inhibitor of Notch signaling Numb. The authors demonstrated that Numb is asymmetrically distributed in the dividing ISC of the enterocyte lineage and that symmetric Numb distribution favors the enteroendocrine fate by inhibiting Notch [33] (Figure 2).

The Dl/Notch pathway, as an actor of lateral inhibition, has also been proposed to control neutral competition resulting from symmetric duplication and loss [26]. If two ISC divide nearby, the four sibling cells might engage Notch-mediated lateral inhibition not between sisters (as seen in the asymmetric division), but between non-sibling cells. One pair of cells remains stem cells (ISC-ISC), while the other becomes precursors (mostly enteroblast-enteroblast), resulting in ISC loss and replacement. This model involving lateral inhibition between non-sibling cells was further refined, providing an alternative, complementary, and likely more frequent model based on the contact area-mediated differential fates of sibling cells [31]. This study demonstrated that the strength of Notch signaling is proportional to the contact area between the two daughter cells: the small contact area (low Notch signaling) mediates ISC-ISC fate, the medium contact area (intermediate Notch signaling) mediates ISC-enteroblast fate, and the long contact area (high Notch signaling) mediates enteroblast-enteroblast fate (Figure 2).

The molecular mechanism controlling the length of the contact zone between the two ISC-derived daughter cells was not described in this work but might involve cell adhesion molecules. During asymmetric division, reducing E-Cadherin in ISC results in an augmentation of the enteroendocrine cell number, which can be explained by a reduction of Notch signaling [38], although this phenotype could not be reproduced [39]. A negative-feedback mechanism seems to take place, whereby a high level of Notch, induced by a strong E-Cadherin-mediated contact, down-regulates the adhesion molecule itself to separate the ISC and the enteroblast [38]. The insulin pathway in enteroblasts might also contribute to the separation of the two daughter cells [39]. All these studies highlight the important role of the Dl/Notch signaling pathway in controlling ISC division modes and precursor fate during gut homeostasis.

### 2.2. Dpp/BMP Signaling Pathway

The Dpp/BMP (Decapentaplegic/Bone Morphogenetic Protein) pathway is another regulator of the ISC division modes [30]. Although different levels of inhibition of Dpp/BMP signaling trigger diverse phenotypes, complete loss alters the outcome of ISC divisions, from mostly asymmetric ISC-enteroblast to predominantly symmetric enteroblast-enteroblast. Conversely, over-activation of the pathway favors ISC-ISC duplication. Thus, Dpp/BMP promotes ISC identity, and it was shown that this action is carried out by down-regulating Notch activity. Sources of the Dpp/BMP signal were first described in the visceral muscles and the tracheal cells [40,41], but the main source for ISC self-renewal is the enterocytes [30]. Enterocyte-expressed ligands Dpp and Gbb (Glass bottom boat) form an apico-basal gradient, which is induced by basement membrane-localized type IV collagens. This model explains the unidirectionality of Notch signaling relative to the basement membrane in the enterocyte and enteroendocrine lineages: the apical daughter cells, located further from the source of Dpp/BMP ligands, are active for Notch signaling, whereas the basal daughter cells, which receive high levels of Dpp/BMP, have no Notch activity (Figure 2).

### 2.3. Polarity Complex and Integrins

From *Drosophila* to mammals, the choice between symmetric or asymmetric division of stem cells is determined by spindle orientation during mitosis [42,43]. In the mouse small intestine, symmetric and asymmetric cell divisions display spindle axes that are parallel or perpendicular to the basal surface, respectively [25,44]. Similarly, in the *Drosophila* intestine, the asymmetric division is guided by an oblique, nearly perpendicular position of the mitotic spindle relative to the basement membrane [13,28]. Apico-basal and planar cell polarities are active processes that align the spindle along a particular axis to influence the division mode in many different cell types and organisms [45]. For the last decade, *Drosophila* neuroblasts have represented a very good model to study the molecular control of cell polarity and cell fate determinants in asymmetric cell division. Each neuroblast divides asymmetrically to give rise to another neuroblast and a smaller ganglion mother cell, which divides one more time to generate two neurons or glial cells. During neuroblast division, the Par complex, consisting of Bazooka (also known as Par-3), Par-6, and atypical Protein Kinase C (aPKC), is localized apically where, together with heterotrimeric G-proteins and the Pins (Partner of inscuteable) complex, it controls the microtubule attachment and mitotic spindle alignment along the apico-basal axis. In *Drosophila* ISC, the asymmetric localization of the Par complex also correlates with asymmetric divisions [13,28] (Figure 2). Whereas loss of function of the Par complex increases the ISC number (symmetric ISC-ISC division), the gain of function triggers enteroblast-enteroblast duplication through activation of the Notch pathway [28].

Loss of function of integrins results in the appearance of ectopic Dl-positive cells, similarly to the Par complex inhibition, but in addition increases the ISC proliferation rate. Interestingly, loss of function of either integrins or the Par complex resulted in reorientation of the mitotic spindle parallel to the basement membrane [28]. These data suggest that integrins, by mediating adhesion of the basally located daughter cell to the basement membrane, provide positional information for the asymmetric segregation of the Par complex to the apical pole, thus promoting the asymmetric division mode (Figure 2). A second study using a viable null allele of the βν integrin subunit confirmed the augmentation of proliferation and of the number of ISC-ISC pairs upon integrin loss of function [46]. These results are, however, in sharp contrast to another study showing that integrin signaling is required for both maintenance and proliferation of ISC, as loss of function (mostly using null alleles) decreases ISC number, as well as their division rate [47]. This discrepancy may result from differences in the alleles used and/or the efficacy of the RNAi lines, and likely reveals complex cell-type or subunit-specific roles of integrins. However, the asymmetric segregation of the Par complex to the apical pole and its role in aligning the spindle perpendicular to the planar plane are similar to what has been described in other tissues and organisms [45]. Goulas and colleagues also observed that integrins are genetically upstream of the Par complex, indicating that they represent, so far, the earliest molecular event in the control of symmetry/asymmetry.

Whereas this last work focused on the enterocyte lineage during asymmetric division, other studies showed that the Par complex, as well as the cell polarity protein Lethal (2) giant larvae (Lgl), also act in the enteroendocrine lineage [13,33]. Indeed, the Par complex promotes Pros localization in basal daughter cells, thus specifying the enteroendocrine precursors [13] (Figure 2). Similarly, the Par complex and Lgl act on Numb, whose symmetric localization in the dividing ISC favors the enteroendocrine lineage by inhibiting Notch activity [33]. These results are very intriguing as they reveal a complex role of the polarity complex. First, its localization at the apical side of the dividing ISC orients the mitotic spindle in the oblique position to trigger the asymmetric division. Second, in the enteroendocrine lineage, it promotes the distribution of Pros in the basally located enteroendocrine precursor and induces the symmetric distribution of Numb (Figure 2). It will be interesting to decipher the molecular mechanism by which the polarity complex can control on one hand asymmetric Pros and on the other hand symmetric Numb. Several other questions also remain unanswered regarding the role of the polarity complex and the integrins, such as: are there any signals further upstream of the integrins controlling asymmetric/symmetric divisions? What is the molecular link between the integrins and the Par complex?

## 3. Division Modes during Adaptive Growth of the Gut

### 3.1. Insulin/Insulin-Like Signaling

A study published in 2011 was the first to describe symmetric divisions in the adult *Drosophila* intestine, revealing a reversible mechanism executed by ISC to promote adaptive growth during gut maturation and in response to fasting/feeding periods [27]. This study demonstrated that feeding-induced growth of the midgut is caused by a strong increase in ISC divisions, both asymmetric and symmetric, but also by a switch towards a predominant symmetric mode of division. Whereas the ratio is 80/20 (80% for the asymmetrical division and 20% for the symmetrical division; see above) during homeostasis, it is reversed during growth (30/70) [27]. The accelerated ISC proliferation and the favored symmetric division therefore allow the gut to produce more differentiated cells, but also to have more ISC, leading to fast tissue growth.

In close contact with ISC that are basally located in the intestinal epithelium, the visceral muscle is well known to influence their behavior [48,49,50,51]. During feeding, the ligand dILP3 (*Drosophila* insulin-like peptide 3), derived locally from the visceral muscle, along with systemic dILP2/5, activates Insulin/Insulin-like (IIS) signaling in ISC, which induces their divisions and favors the symmetric mode [27] (Figure 3a). IIS signaling therefore drives organ growth, a process that is reversibly regulated by dietary changes. During nutrient deprivation, decreased IIS activity results in reduced ISC proliferation due to prolonged contact between ISC and precursors [39]. The micro-RNA *miR-305*, which targets components of both IIS and Notch signaling pathways, has been shown to coordinate the nutritional adaptive growth [29]. On a poor diet, the *miR-305* level is high in ISC, where it targets and represses the Notch signaling repressor Hairless, resulting in upregulation of Notch-mediated transcription, and consequently, increased asymmetric divisions (Figure 3b). On a rich diet, *miR-305* is transcriptionally repressed by IIS signaling. Hairless expression is therefore stabilized and can down-regulate Notch target gene transcription, which favors ISC-ISC duplication and organ growth. Another actor, the RNA-binding protein Lin-28, also promotes ISC symmetric division in fed flies by interacting with the insulin receptor (InR) mRNA, increasing its expression and enhancing IIS signaling [52,53]. On a poor diet, the InR level within ISC is post-transcriptionally down-regulated as a result of Lin-28 inhibition by the RNA-binding protein Fragile X Mental Retardation Protein (FMRP) [53] (Figure 3a,b).

The mouse intestine is also known to respond to dietary signals [54,55,56,57,58,59]. Surprisingly, however, calorie restriction induces an expansion of the mouse Lgr5 + ISC pool, accompanied by a reduction of mature enterocytes [54,56]. The insulin/mTORC1 (mammalian target of rapamycin complex 1) pathway is down-regulated in Paneth cells, which allows the activation of LKB1/AMPK/SIRT1 (Liver kinase B1/AMP-activated protein kinase/Sirtuin 1) signaling cascade in ISC through the paracrine factor cyclic adenosine diphosphate ribose [54,56]. In parallel, mTORC1 is upregulated in ISC (independently of insulin), where it collaborates with SIRT1 to foster ISC self-renewal [56]. On the contrary, two other studies showed that Lgr5 + ISC number did not change upon fasting, but that instead, the quiescent ISC +4 contributed to intestinal growth after calorie restriction via the insulin/mTORC1 pathway [58,59]. A high-fat diet also drives ISC hyperproliferation, and this process requires insulin and the activation of the PPAR-δ (peroxisome proliferator-activated receptor delta) pathway [55,57]. These studies did not analyze the cellular mechanisms driving ISC pool expansion, but it was later shown that the ISC of fasted mice exhibit a switch in favor of asymmetric divisions and that a high-glucose treatment of organoids does the opposite (more symmetry at the expanse of asymmetry) through a reduction of the LKB1/AMPK pathway [44].

All these studies performed in *Drosophila* and mice indicate that the insulin pathway is central to the ISC dietary response, but also that other signaling events play a role in the adaptive growth of the mouse small intestine (Table 1). Future studies will have to determine whether these other pathways operate in the *Drosophila* midgut to regulate the ISC division modes, and it will also be important to analyze whether nutrient quality (lipids, proteins, carbohydrates) could induce different responses.

### 3.2. Mitotic Spindle Orientation via JNK Signaling

Several studies noticed that ISC division modes are linked to the orientation of the axis between the two daughter cells relative to the plane of the epithelium [13,28,33,36,37]. Interestingly, live imaging showed that horizontal-vertical reorientations can occur during the time of mitosis itself [60]. Recently, a detailed analysis revealed that the switch between symmetric and asymmetric stem cell divisions in the adult *Drosophila* intestine requires a change in the alignment of the mitotic spindle [61]. This study showed that the symmetric division is driven by a planar (≤15°) spindle, whose orientation is controlled by c-Jun-N-terminal kinase (JNK) signaling acting at two levels [61] (Figure 3c,d). First, the activated JNK/Basket, which is associated with the spindle, directly controls the recruitment of the centrosome-associated protein Wdr62 (WD repeat domain 62) (Figure 3c). Second, JNK signaling mediates transcriptional repression of the kinesin Kif1A (kinesin family member 1A). In absence of the JNK signal, Wdr62 is localized to the centrosomes, whereas Kif1A could orient astral microtubules to the cell cortex, the two proteins interacting with cortical determinants of spindle orientation, such as Pins or Mud (Mushroom body defect), in order to favor the oblique position of the spindle (Figure 3d). JNK-induced phosphorylation of Wdr62 and repression of Kif1A therefore interfere with this process and promote together the planar alignment of the mitotic spindle. This molecular action favors symmetric division during diet-induced adaptive growth, but also during stress response and aging (see below). In mice, we described above that it is rather the LKB1-AMPK pathway that controls the nutrient-driven switch between the two division modes [44], but it is not known whether it acts directly or indirectly on the mitotic spindle and whether the JNK pathway is involved as in *Drosophila* (Table 1).

## 4. Stress and Aging on ISC Division Modes

### 4.1. Stress-Modulated Switch between Symmetric and Asymmetric Divisions

The symmetric/asymmetric division ratio allows the epithelium to adapt to the nutrient needs, as seen above, but also to face stressful and damaging conditions. In mice, when normal homeostasis is disrupted by inflammation or oncogenic mutations, excessive proliferation is compensated by a switch in favor of asymmetric division to restrain the number of Lgr5 + ISC [62]. This is achieved through a feedforward loop involving *miR-34a*, Numb, and Notch (Table 1). In *Drosophila*, the JNK signaling pathway promotes the symmetric outcome of ISC divisions during intestinal growth after refeeding, as discussed above, but also after paraquat treatment [61] (Figure 4). However, bacterial stress with the gram-negative *Erwinia carotovora carotovora 15* (*Ecc15*) bacterium has no effect on spindle orientation, suggesting that not all the stresses can realign the spindle [61]. This is, however, in contradiction with another study, showing that pairs of Dl-positive cells increase upon *Ecc15* infection, likely reflecting a shift towards ISC symmetric divisions [63]. In the same way, the DNA-damaging agent bleomycin or infection with another gram-negative bacterium, *Pseudomonas entomophila*, drives ISC expansion by promoting their symmetric division, but not the chemical dextran sulfate sodium that increases only ISC proliferation rate [64]. Moreover, ISC partial loss does not trigger any compensatory mechanism (increased division rate and/or mode) to recover the ISC pool [65]. These studies therefore indicate that the stress response does not systematically trigger the symmetric/asymmetric division switch.

In the study of Zhai and colleagues (2017), *Ecc15* infection favors the ratio of Dl+/Dl+ pairs (ISC-ISC) to the detriment of the ratio of Dl+/Notch+ pairs (ISC-enteroblast). In addition, strong cell contact mediated by the cell adhesion molecules E-Cadherin and Connectin ensures a high level of Notch signaling in one of the daughter cells. Acting in parallel to Notch, a signaling cascade of JAK/STAT-Sox21a-GATAe-Dpp/BMP cooperates to accelerate the differentiation step between the enteroblast and the enterocyte. These two mechanisms therefore seem to trigger an increase in symmetric divisions and a fast production of enterocytes, leading to a rapid recovery of the intestine upon aggression. It is interesting to note, however, that this study also showed that Dpp/BMP signaling, GATAe, and E-cadherin/Connectin are not critical for enteroblast differentiation during homeostasis.

Stress induced by bleomycin feeding or bacterial infection increases the expression of the ligands Dpp and Gbb in enterocytes, resulting in a high level of Dpp/BMP signaling activity in both daughter cells [64] (Figure 4). As the Dpp/BMP pathway antagonizes Notch signaling, which is important for enteroblast specification, the two daughter cells keep their stem cell state, leading to an expansion of the ISC pool size [64]. Therefore, Dpp/BMP signaling does not directly act on the division mode, but rather instructs cell identity, and it is likely that JNK signaling is also activated to orient the mitotic spindle parallel to the basement membrane. Interestingly, a high level of BMP activity in enterocytes also triggers the downregulation of *dpp* and *gbb* gene expression. This negative feedback loop in enterocytes is important to reduce ISC population size (by favoring enteroblast-enteroblast symmetric duplication) and to restore homeostasis after tissue regeneration.

What could be the stress-induced signal initiating the symmetric/asymmetric division switch? The JNK pathway is a good candidate to fulfill this function, as (i) it directly controls the spindle orientation, (ii) it is a notorious stress response pathway, and (iii) it has been described as an activator of *dpp* expression [61,66] (Figure 4). The level and/or the timing of JNK activation might be the signal that triggers the switch. For example, mild stress would cause only a small, transient, JNK response, leading to an increased division rate with conservation of the asymmetric mode to replace the affected differentiated cells. In response to robust stress, more ISC are required and could be obtained by high, sustained, JNK activity, which switches the mode in favor of symmetric divisions both by controlling the orientation of the mitotic spindle (through Wdr62 and Kif1A) and by inducing expression of the Dpp/BMP ligands. This model would explain the discrepancies obtained with *Ecc15* infection, which could trigger different responses according to the experimental procedures. In support of this model, it has been shown that a low amount of opportunistic bacteria induces mild early stress response and JNK activation, whereas a high level triggers a prolonged response [67]. Further work is required to definitely decipher the molecular mechanisms involved in stress-induced symmetric/asymmetric ISC division modes and the role of the JNK pathway in this process.

### 4.2. ISC Division Fate during Aging

In both mammals and insects, aging of the intestine involves microbial dysbiosis, barrier dysfunction, and inflammation. However, whereas studies in mice, though still conflictual, suggest that ISC proliferate less with age [68,69,70,71], studies in *Drosophila* have clearly shown that ISC accelerate their proliferation rate [69,72,73,74,75]. In old mice, a reduced capacity of the Lgr5+ ISC to proliferate is a consequence of increased mTORC1 signaling, which triggers the decline of Wnt/Wg signaling through the production of the extracellular Wnt inhibitor Notum [68,70] (Table 1). In aged *Drosophila*, loss of acidity in the lumen, induced by metaplasia in the acidic region of the midgut, and immuno-senescence provoke dysbiosis, which in turn triggers chronic elevation of ROS (reactive oxygen species) level in a DUOX- and Keap1/Nrf2-dependent manner [72,74,75,76,77]. High ROS level then activates several signaling pathways, among those JNK [72,73,74]. These signals increase ISC proliferation, but also lead to their mis-differentiation, promoting general gut dysplasia (Figure 5). The progressive loss of cell junctions in enterocytes also contributes to dysplasia formation in a JNK-dependent manner and reduces gut integrity [78,79]. A previous study showed that loss of the βν integrin subunit in enterocytes leads to premature aging of the gut, along with increased JNK activation and ISC symmetric divisions [46]. In addition, in aged flies, the majority of mitotic spindles in dividing ISC are in the planar plane, which is due to high JNK signaling [61]. Restoring the apico-basal orientation of the spindle in aged flies by manipulating the level of JNK, Wdr62, or Kif1A promotes the asymmetric cell fate, delays loss of barrier function, and extends lifespan. All these results indicate that JNK activity increases with age in an ROS-dependent manner, both in enterocytes and in ISC and, along with other signaling pathways, promotes ISC proliferation and mis-differentiation (Figure 5). In addition, JNK in ISC induces the alignment of mitotic spindles in the planar plane, which makes ISC symmetric division a hallmark of aging contributing to the loss of tissue homeostasis in old animals.

## 5. Conclusions and Outlooks

In *Drosophila*, ISC division mode, driving two stem cells (symmetric) or one stem cell and one precursor (asymmetric), is a key component in adapting the needs of the gut, whether during homeostasis, adaptive growth, stress-induced regeneration, or aging. The process leading to asymmetric division originates at the cell membrane and is mediated by integrins and polarity complexes, which promote an oblique orientation of the mitotic spindle with respect to the basement membrane. The Dl/Notch and Dpp/BMP signaling pathways then act to instruct a specific cell fate in each of the daughter cells. During gut homeostasis, 80% of the divisions are asymmetric, and the remaining 20% are symmetric. In the latter case, JNK signaling counteracts the cortical polarity complexes to re-orient the mitotic spindle parallel to the basement membrane. How the precise repartition of the two division modes is maintained is an important, still open, question, but could rely on the Dl/Notch-mediated lateral inhibition. In response to environmental cues, ISC increase their rate of symmetric divisions. This is driven by the IIS and JNK pathways during nutrient-induced growth of the intestine, and the JNK and Dpp/BMP pathways after bacterial or chemical-induced damages. The molecular connections between JNK and the other pathways remain to be established in each condition. During aging, JNK contributes to dysplasia by forcing ISC to increase their symmetric division rate. All these data indicate that JNK signaling is a core molecular determinant of fly gut adaptation through its direct effect on spindle orientation (Table 1). In the mouse small intestine, switching of ISC division modes has been linked to nutrient-induced growth, inflammation, and tumorigenesis. The signaling pathways are starting to be well described (Table 1), but in general, fewer detailed studies on the molecular mechanisms have been carried out. It will be interesting to analyze whether JNK signaling could also be an important regulator of symmetric/asymmetric division modes in mice. In turn, it would also certainly be beneficial to examine in *Drosophila* if the pathways described in the mouse could be involved in the symmetric/asymmetric switch of ISC divisions. Further study of the mechanisms controlling the modes of ISC division is crucial to better understand how the intestine, and tissues in general, adapt to extrinsic signals, as well as how their deregulation can lead to the appearance of pathologies, in particular dysplasia and cancer.

## Figures and Tables

**Figure 1 ijms-21-06362-f001:**
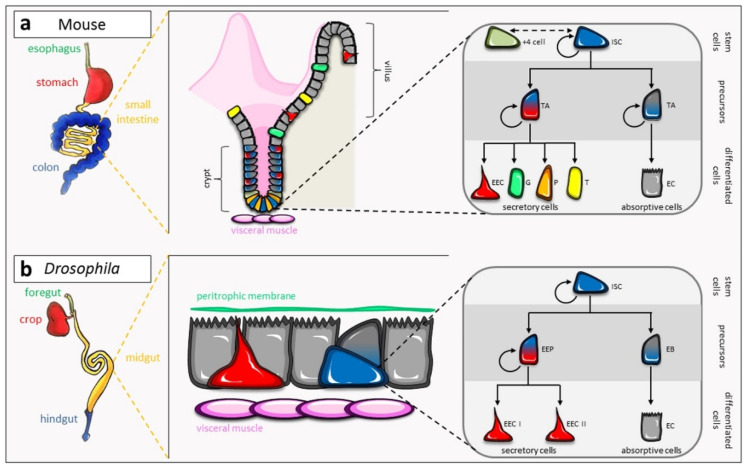
Cellular organization and lineages of the mouse small intestine and the *Drosophila* midgut. (**a**) The small intestine of the mouse is organized as units made of crypts and villi. ISC, localized at the bottom of the crypts, and transit-amplifying precursors (TA) proliferate to generate mature cells (enteroendocrine cells—EEC, goblet cells—G, Paneth cells—P, tuft cells—T, enterocytes—EC) through two distinct lineages, forming the secretory cells and the absorptive cells. Whereas Paneth cells migrate to the bottom of the crypt, the other differentiated cells migrate up to the villi. (**b**) The midgut of *Drosophila* is a flat epithelium. Two types of precursors are generated from ISC divisions, the enteroendocrine precursors (EEP) and the enteroblasts (EB), which give rise to the secretory enteroendocrine cells (class I and class II) and the enterocytes, respectively. The peritrophic membrane has an equivalent role as the mucus of the small intestine. See the main text for details.

**Figure 2 ijms-21-06362-f002:**
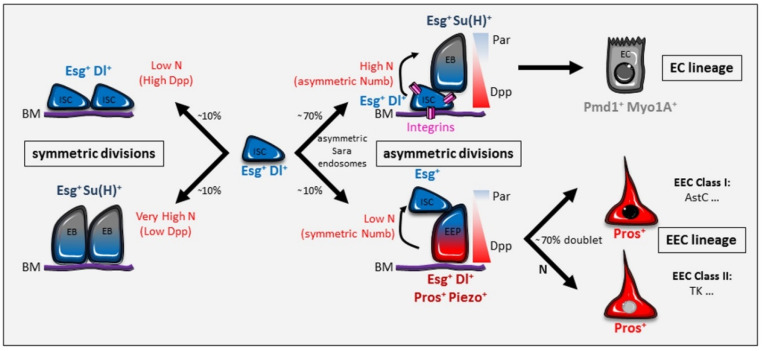
Division modes of *Drosophila* ISC. The interplay between Notch and Dpp/BMP signaling pathways, as well as the Par complex, integrins, Numb, and Sara endosomes, regulates the different daughter cell fates of ISC divisions: ISC-enteroblast, ISC-enteroendocrine precursor, ISC-ISC, and enteroblast-enteroblast. Symmetric division of the ISC can also give rise to two enteroendocrine precursors, but the signaling events have not been described (and therefore are not included in the figure). The localization of the Par complex in the apical daughter cell is necessary for the ISC asymmetric division. Two lineages, that of enterocytes and that of enteroendocrine cells, are produced depending on the level of Notch activity, which is itself regulated by Dpp/BMP, Numb, and Sara endosomes. In addition, in the enteroendocrine lineage, the Par complex regulates the correct localization of Pros in the basally located enteroendocrine precursor. Integrins allow the asymmetric localization of the Par complex during ISC-enteroblast asymmetric division, but their specific role in the enteroendocrine lineage is not known. Two main classes of enteroendocrine cells, expressing Allatostatin C (AstC) for class I and Tachykinin (Tk) for class II, are produced from the enteroendocrine precursors. Around 70% of the enteroendocrine cells form a doublet coming from the direct division of the enteroendocrine precursors. During the symmetric division, low Notch signaling (due to high Dpp/BMP) leads to ISC-ISC production. Conversely, a high Notch (low Dpp/BMP) leads to enteroblast-enteroblast production. Percentages are related to the homeostatic condition. Each cell type of the midgut can be distinguished by specific markers. See the main text for details. BM: basement membrane; ISC: intestinal stem cell; EB: enteroblast; EEP: enteroendocrine precursor; EC: enterocyte; EEC: enteroendocrine cell; N: Notch.

**Figure 3 ijms-21-06362-f003:**
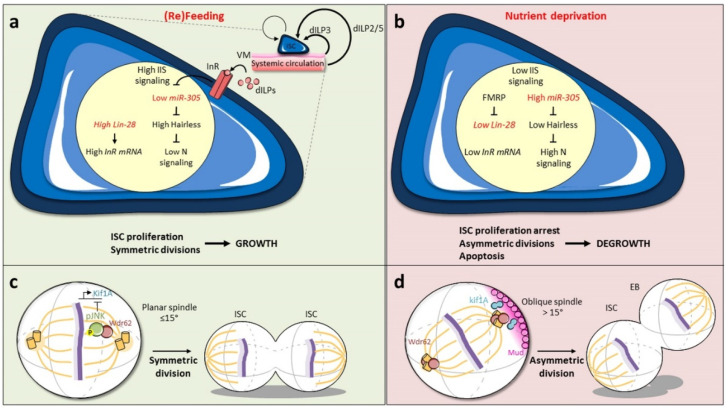
Adaptive growth of the gut depends on nutrient availability. Feeding (**a**) and nutrient deprivation (**b**) impact the ratio of symmetric and asymmetric divisions to mediate the adaptive growth of the gut through insulin signaling. (**c**,**d**) JNK signaling counteracts cortical determinants to regulate the planar orientation of the mitotic spindle and promotes the symmetric division. See the main text for details.

**Figure 4 ijms-21-06362-f004:**
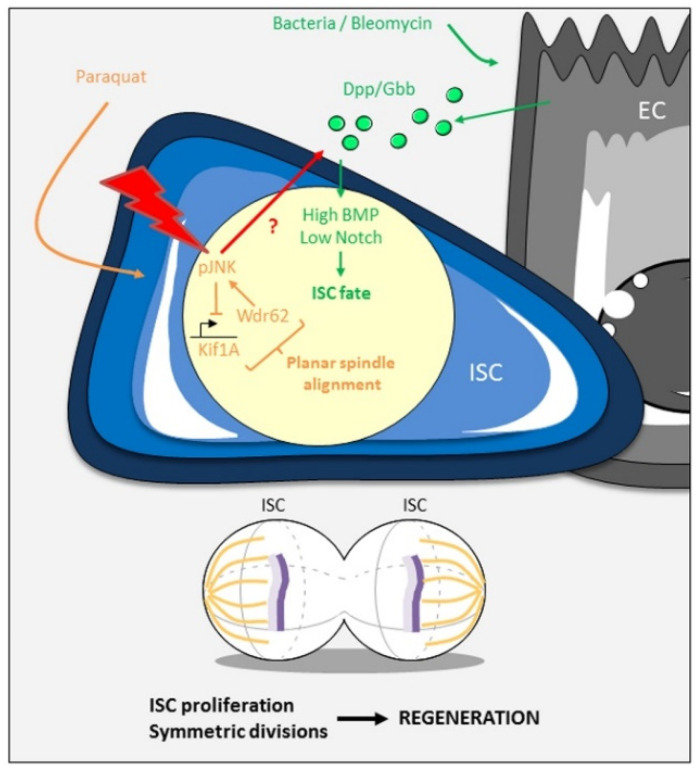
Stress-response of the gut by increased symmetric divisions. JNK-induced orientation of the mitotic spindle, Dpp/BMP-driven ISC fate and fast enteroblast differentiation through strong adhesion allow the gut to produce more ISC and differentiated cells to compensate for the loss of damaged cells. JNK signaling might be the central stress sensor (red flash). Activated JNK can then act directly on the planar spindle alignment and also could induce the expression of the Dpp/BMP ligands (red arrow). See the main text for details.

**Figure 5 ijms-21-06362-f005:**
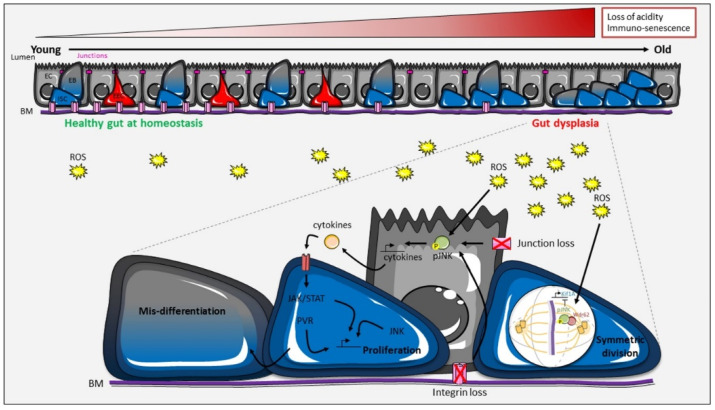
An increase in symmetrical divisions contributes to the aging of the *Drosophila* intestine. Aging is accompanied by loss of lumen acidity and epithelial immuno-senescence, leading to dysbiosis. A rise of reactive oxygen species (ROS) level, as well as the progressive disappearance of cellular junctions and integrins, trigger the activation of several signaling pathways, including JNK. JNK participates in hyper-proliferation of the ISC and increases symmetric divisions by orienting the mitotic spindle parallel to the basement membrane (BM). See the main text for details.

**Table 1 ijms-21-06362-t001:** Summary of the signaling pathways and their regulators (when known) involved in the gut responses to adaptive growth, stress, and aging both in *Drosophila* and the mouse. The pathways regulating the orientation of the mitotic spindle are also indicated. However, the direct molecular role of the LKB1/AMPK pathway in mouse ISC has not been proven (*). See the main text for details.

Process	*Drosophila* Midgut	Mouse Small Intestine
Adaptive growth	Insulin/IIS signaling Notch JNK	Insulin/mTORC1 LKB1/AMPK/SIRT1 Cyclic adenosine diphosphate ribose PPAR-δ
	Regulators: *mir-305*, Lin-28, FMRP	
Stress	Notch Dpp/BMP JNK	Notch
		Regulators: *mir-34a*, Numb
Aging	JAK/STAT PVR JNK	mTORC1 Wnt/Wg
		Regulators: Notum
Mitotic Spindle	JNK	LKB1/AMPK *

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
