# Peer review of "Tissue Adaptation to Environmental Cues by Symmetric and Asymmetric Division Modes of Intestinal Stem Cells"

_ijms, 2020, doi:10.3390/ijms21176362_

Round 1

Reviewer 1 Report

In this review, Joly and Rousset address the role of symmetric versus asymmetric cell division in the development, homeostasis, and response to stress of the gut/intestine. Specifically, they review what it known about the mechanisms controlling these processes in the adult Drosophila midgut and compare those to what is known for the mouse small intestine. The review focuses on physiological and molecular events mainly affecting major signaling pathways and cortical proteins known to influence mitotic spindle position and the resulting symmetric or asymmetric division, and how these divisions affect cell identity and tissue homeostasis.

Overall the topics addressed in this manuscript will be of interest to a wide audience, and this review will be a valuable contribution to the literature. In its current form the manuscript can difficult to follow in some sections, which will likely limit the impact of the publication beyond experts who are already familiar with the topics. Therefore, this Reviewer believes the manuscript will benefit from revisions prior to publication. The comments below are largely related to this issue.

Introduction:

The first paragraph introduces terms such as crypt and villi. It states that cells become transit-amplifying precursors that can still divide. Absorptive precursors then differentiate into EC, while secretory precursors generate several cell types. Readers who are less familiar with the topic may be left to infer whether absorptive and secretary precursors arise directly from transit-amplifying precursors, and what steps are in-between. In general, there are many cells types introduced but little context to orient readers to the overall process/organization.

The introduction may benefit from an additional figure that orients readers to tissue organization and cell populations to provide an aid for adding context to the text. This may help readers digest the several cells types mentioned and compared in the introduction and throughout the manuscript.

At the end of second paragraph it’s stated, “Interestingly, five Drosophila cell types…part of the EC lineage.” This statement should have a citation.

Section 2:

The biggest concern for this Reviewer is the number of abbreviations used in the manuscript together with the dense text lacking sufficient context in some areas. In general, there are too many abbreviations used for readers less familiar with the subjects. In section 2 this is exasperated by the dense text describing a number of cell divisions and alternative outcomes. Adding to this challenge is the long length of some paragraphs presenting a string of details (and sometimes alternatives). For instance, the paragraph beginning on line 197 is nearly a page in length. While the language is clearly written and presented sequentially, it is a lot to process in one chunk, and the abbreviations make it less accessible to process while reading. The result is that the information seems to lack context and, honestly speaking, becomes difficult to follow clearly. (This comment is addressed to other sections also, but Section 2 highlights the concern.)

Section 3:

The scope of the review is presented as what is known about gut development in Drosophila in parallel to what is known in mammalian small intestine (mouse). In sections 3.1 and 3.2 the mouse information/comparison is essentially just a sentence at the end. In section 3.1 it is stated that nutrient availability drives the mode of division like in Drosophila, thus demonstrating the process is conserved throughout evolution. This is quite vague and leaves a reader to wonder if all the details described for fly are the same in mouse, or simply the correlation between nutrient and mode of division? What about the players described in the preceding paragraphs and in Figure 2?

Similarly, in section 3.2 after describing the situation in Drosophila it is stated that rather than the JNK pathway, it is the LKB1-AMPK pathway that controls spindle orientation in mice. It would be helpful for the reader to clarify whether the other details described in fly are conserved, are orchestrated by other players, or unknown in mouse. This would more nicely complete the stated objective described as what is known about gut development in Drosophila in parallel to what is known in mammalian small intestine. (Some additional helpful statements are made in the conclusion section with this regards, but they come too late with respect to the information presented in Section 3).

Section 4:

Section 4 comes across to the reader much better. For instance, Section 4.1 does a nice job by providing details about various processes and then placing them in the context of the other processes and overall events being discussed. Thus, it is much easier for the reader to follow, and absorb information while reading than the preceding sections mentioned above.  

Minor comments:

Line 77 - The sentence starting with ‘After asymmetric division, the resulting EEP can directly…”:  The beginning of this sentence seems unclear following the context of the proceeding sentences. After asymmetric division of what?

The paragraph starting on line 283 addressing Figure 2c and 2d:  The text clearly describes the case in panel 2c in which JNK recruits Wdr62 and represses Kif1A. However, the situation in 2d is left to be implied as the opposite case, which may be somewhat obscure, especially since Wrd62 is also shown in this panel, but its function here is not addressed. Perhaps a sentence or two describing this panel would be helpful.

Line 64: Should ‘Ligand’ be capitalized?

Line 112 - syntax: Perhaps it should read as “The mechanisms involved in this regulation have started to be better described…complete understanding of the molecular processes….” Also, the end should likely have a qualifier such as “adult stem cells of the intestine”.

Sentence on line 371 may read better as: “…the majority of mitotic spindles in dividing ISC are in the planar plane, which…”

Sentence on line 393 may read better as: “During gut homeostasis, 80% of the divisions are asymmetric, and the remaining 20% are symmetric.”

Reviewer 2 Report

The review by Joly and Rousset outlines the current comprehension of the mechanisms regulating intestinal stem cell division modes (asymmetric versus symmetric) in Drosophila midgut (a well-recognized model to study tissue renewal and plasticity) in parallel to the current knowledge in the mammalian small intestine. Authors decipher the signalling pathways in the gut involved during homeostasis, adaptive growth, and during the response to stress and ageing. The review is well organized, very well written, and cites appropriate and up to date references. The authors were able to deal with a complex topic by exposing it in a clear, logical and sequential way. Figures are nice and adequately summarize the critical aspects detailed in the text. The review offers a complete narration of what is known in the literature, which is, at the same time, critically discussed and commented by the authors (much appreciated).

I think that the review will be of interest for the readers of IJMS and I endorse its publication in the present form.

Reviewer 3 Report

IJMS - 893886

Tissue adaptation to environmental cues by symmetric and asymmetric division modes of intestinal stem cells Aurélia Joly and Raphaël Rousset

Summary

In this manuscript, Joly and Rousset reviewed past and present observations linked to the stem cell division modes in Drosophila. While the content related to Drosophila midgut seems to be well investigated, I miss basically the comparisons to mammals promised in the abstract (l. 18). Although the introduction to the main text provides sufficient information on both, mice and fly, the subsequent sections are focused on Drosophila research only.

Considering the fact that I’m not an expert in this field, I’m worried about the following issues:

1) The abstract and title are misleading. Beside the point that there are too little comparisons in the main text between mammals and fly, some parts of the abstract gives the impression, that the authors present their own research, for instance ‘We decipher’ (l. 19) or ‘JNK signaling appears’ (l. 22) or ‘Studying the mechanisms’ (l. 23). Hence the abstract should be revised. The title is misleading as well, as the entire review is focused on Drosophila midgut, a single tissue. For instance, FlyAtlas distinguishes more than 20 tissues in Drosophila.

2) Maybe I missed this point, but I was not able to find any information about other mammals than mice in the main text. It is definitely okay to compare mouse small intestine and fly midgut only, but in this case, mammals is misleading.

3) Any figure presented in this manuscript is related to Drosophila. Differences/Agreements with mammals/mice or similar pathways should be visualized too.

4) The JNK pathway, explicitly highlighted in the abstract, has been revised recently for fly, with slightly different content:

Two-Faced: Roles of JNK Signalling During Tumourigenesis in the Drosophila Model

John E. La Marca* and Helena E. Richardson

Pro-apoptotic and pro-proliferation functions of the JNK pathway of Drosophila: roles in cell competition, tumorigenesis and regeneration

Noelia Pinal, Manuel Calleja, and Ginés Morata

What is the connection/difference between the JNK information presented in these reviews and the current manuscript?

5) The comparison to mammalian/mice must be improved. I would suggest to provide figures visualizing the analogous mechanics in mammals or tables. Further data on mice, stem cell division or tracing lineage experiments can be found, for instance in the following publications:

a) (mouse epidermis, not intestine)

Spatiotemporal coordination of stem cell commitment during epidermal homeostasis

Panteleimon Rompolas, Kailin R. Mesa, Kyogo Kawaguchi, Sangbum Park, David Gonzalez, Samara Brown, Jonathan Boucher, Allon M. Klein, Valentina Greco

b) (mouse model and experimental work - small intestine)

Stem cell competition in the gut: insights from multi-scale computational modelling

Torsten Thalheim, Peter Buske, Jens Przybilla, Karen Rother, Markus Loeffler, Joerg Galle

Intestinal crypt homeostasis revealed at single-stem-cell level by in vivo live imaging.

Ritsma L, Ellenbroek SI, Zomer A, Snippert HJ, de Sauvage FJ, Simons BD, Clevers H, van Rheenen J.

Redundant sources of Wnt regulate intestinal stem cells and promote formation of PCs

Farin HF, Van Es JH, Clevers H.

Regulation and plasticity of intestinal stem cells during homeostasis and regeneration

Joep Beumer, Hans Clevers

6) Does the fly stem stem cell division follows one of the concepts available in mice?

Cell Organisation in the Colonic Crypt: A Theoretical Comparison of the Pedigree and Niche Concepts

Richard C. van der Wath , Bruce S. Gardiner, Antony W. Burgess, David W. Smith

Minor issues

  1. lines 6-9. The affiliation of both authors seems to be the same, but each author has his own tag (Joly – 1, Rousset – 2) for this affiliation.

  2. Many data is presented in the text. Maybe, some of these information can be comprised as a table and hence, improve its legibility.

  3. Abbreviation Notch (N) (l. 64): The acronym N is linked to many terms and thus makes it confusing to the reader. I am not aware, if this abbreviation is used frequently in the authors community, for readers from other disciplines it might be helpful to use “Notch” instead of N or use a more distinct abbreviation.

  4. The authors rise the question, what is initiating symmetric/asymmetric division (l. 343). Are there any other hypotheses beside the JNK pathway?

  5. Even though I agree to the statement “ISC proliferates different less with age in mice” (l. 360), I miss some references here.

  6. Are there any publications reporting the accelerated proliferation rate in Drosophila intestine (l. 361)?

  7. The conclusions contain a summary on fly only.

Round 2

Reviewer 3 Report

The manuscript has been substantially improved. I'm fine with this version.

One comment: The reviewer portal provides a "Supplementary file", which seems to be a previous version of this manuscript. Please check with the editors